# Effects of Chronic Mild Stress on Cardiac Autonomic Activity, Cardiac Structure and Renin–Angiotensin–Aldosterone System in Male Rats

**DOI:** 10.3390/vetsci9100539

**Published:** 2022-09-29

**Authors:** Janpen Bangsumruaj, Anusak Kijtawornrat, Sarinee Kalandakanond-Thongsong

**Affiliations:** 1Interdisciplinary Program in Physiology, Graduate School, Chulalongkorn University, Bangkok 10330, Thailand; 2Department of Veterinary Physiology, Faculty of Veterinary Science, Chulalongkorn University, Bangkok 10330, Thailand

**Keywords:** cardiac autonomic balance, cardiac structure, chronic mild stress, heart rate variability, renin–angiotensin–aldosterone system, angiotensin 1 receptor, rat

## Abstract

**Simple Summary:**

Stress can cause heart disease by changing cardiac structure and/or function as assessed by heart rate variability; however, this may or may not happen when exposed to daily stress. Therefore, this study was conducted to investigate alterations in cardiac structure and cardiac control through autonomic activity, and in the renin–angiotensin–aldosterone system in male rats after exposure to 4-week chronic mild stress as an animal model for daily stress exposure in humans. The results showed that chronic mild stress increased both sympathetic and parasympathetic autonomic activities without affecting their balance. The renin–angiotensin–aldosterone system, which was partly responsible for changes in cardiac structure, was also affected. These data suggest that cardiac autonomic control and cardiac structure were affected by 4-week mild stress; however, this effect was probably in an adaptable period as the sympathetic and parasympathetic activities were in balance.

**Abstract:**

Stress is associated with cardiovascular disease. One accepted mechanism is autonomic imbalance. In this study, we investigated the effects of chronic mild stress (CMS) on cardiac autonomic control, cardiac structure and renin–angiotensin–aldosterone system (RAAS) activity in adult male Sprague Dawley rats. The CMS model provides a more realistic simulation of daily stress. The animals were divided into control and CMS, and were exposed to 4-week mild stressors. The electrocardiogram recording, sucrose intake and parameters related to stress, cardiac alterations and RAAS were determined. The results showed that CMS had lower body weight and higher sucrose intake. The heart rate variability (HRV) revealed that CMS increased autonomic activity without affecting its balance. The increased RAAS activity with upregulated angiotensin type 1 receptor mRNA expression was shown in CMS. The increased sympathetic activity or RAAS was correlated with stress. Moreover, the altered cardiac structure (i.e., heart weight and cardiomyocyte cross-sectional area) were correlated with stress-, sympathetic- and RAAS-related parameters. These indicated that CMS-induced cardiac hypertrophy was the result of both sympathetic and RAAS activation. Therefore, it could be concluded that 4-week CMS in male rats induced negative emotion as shown by increased sucrose intake, and increased cardiac autonomic and RAAS activities, which may be responsible for mild cardiac hypertrophy. The cardiac hypertrophy herein was possibly in an adaptive, not pathological, stage, and the cardiac autonomic function was preserved as the autonomic activities were in balance.

## 1. Introduction

It has long been known that stress is associated with cardiovascular disease [1], which is the number one cause of death in the world [2]. In 2019, cardiovascular diseases (ischemic heart disease and strokes) accounted for 27% of the world’s total deaths [2]. There is evidence linking long-term stress with cardiovascular diseases such as hypertension, coronary heart disease, arrhythmias and sudden cardiac death [3,4,5,6]; for example, longitudinal surveys in Australian adults revealed that personal stress or family-related stress increased the likelihood of the onset of heart and/or circulatory diseases [7]. In experimental animal models, it was shown that in a rat model of pressure overload, the cardiac structure (i.e., left ventricular wall thickness and mass, and fibrotic index) and functions (i.e., ejection fraction and hypertension) were worsened by chronic restraint stress [8]. Similarly, in mice, chronic psychosocial stress-induced cardiac structural alteration, i.e., myocardial fibrosis of the left ventricular myocardium [9]. As one of the underlying mechanisms involving cardiovascular disease in humans, cardiovascular autonomic imbalance could be induced by enhanced sympathetic tone, decreased parasympathetic tone or both [10]. Alterations in this cardiovascular autonomic balance can be assessed with heart rate variability (HRV), in which both time and frequency domains have been used successfully to reflect sympathetic and parasympathetic influences [11]. Clinically, the reduction in HRV has been associated with various stressful situations [11]. Similarly, in various animal stress models, the changes in HRV parameters have been demonstrated in rodents subjected to different types of stress such as acute 15-min restraint stress [12], chronic psychosocial stress [9], chronic social defeat stress [13] and 2–4 weeks of chronic mild stress [14,15,16,17]. Despite the availability of different animal stress models, the chronic mild stress (CMS) model is of interest as it utilizes various types of mild stressors, providing a more realistic simulation of stressors in daily life [18]. Moreover, it is known that being exposed to the same stressor repeatedly, as in the repeated-stressors regimen, can lead to the habituation of the hypothalamic–pituitary–adrenal (HPA) axis [19]. On the contrary, the CMS model is different, as it is based on various types of mild stressors, and the HPA axis response to each novel stressor is thus likely to remain intact [19]. Therefore, it is interesting to determine whether CMS can alter the cardiac autonomic activity or cardiac structure (i.e., cardiac hypertrophy), and thus contribute to stress-related cardiac diseases.

Cardiac hypertrophy is one of the cardiac alterations that occurs as an adaptive response to improve cardiac function. However, cardiac hypertrophic responses should not always be considered as a compensatory mechanism to improve cardiac function; maladaptive responses such as in heart failure or contractile dysfunction should also be considered [20]. Through clinical, experimental and animal studies, neurohormones such as epinephrine and norepinephrine (NE) of the sympathetic system, and angiotensin II (Ang II) and aldosterone of the renin–angiotensin–aldosterone system (RAAS) have been identified as the most important neurohormones stimulating pathological cardiac hypertrophy [21,22,23], particularly the RAAS. The increased activity of the RAAS is associated with cardiovascular diseases and its associated damage to organs including the heart [24]. It is known that the RAAS has two pathways with opposite effects: the classic angiotensin-converting enzyme (ACE)/Ang II/Angiotensin type 1 receptor (AT1R) and the new ACE2/Ang 1–7/ Mas1R and Ang type 2 receptor (AT2R) pathways [25]. The former pathway is responsible for hypertension, ventricular hypertrophy, myocardial infarction and atherosclerosis, while the latter has protective effects [24,25]. Therefore, an imbalance of these two pathways is critical in the development of cardiovascular disease [24]. In terms of RAAS activity and HRV, it was demonstrated that the very low frequency (VLF) of HRV was higher upon ACE inhibitor administration [26]. In addition to the new RAAS pathway, pathological cardiac hypertrophy could also protect via the action of atrial natriuretic peptide (ANP) and brain natriuretic peptide (BNP) [27]. Utilizing specific antagonists and knockout studies, it was demonstrated that ANP and BNP played a counter-regulatory role against the fibrosis intracardiac RAAS and may eventually prevent cardiac hypertrophy [27]. It has therefore been used clinically and experimentally as diagnostic and prognostic markers for hypertrophy and heart failure, and as biomarkers in biomedical research to assess hypertrophic response in cell culture and animal models [28]. Moreover, it was shown that ACE2 can regulate ANP production through Ang 1–7 [25]. ANP and BNP have also been associated with stress: in rodents subjected to stress such as immobilized or social stress, the ANP and/or BNP were increased and were abolished through adrenoceptor blockades [29,30,31].

Overall, despite the numerous reports mentioned above, the effects of CMS that represent daily stress in humans on cardiac autonomic activity, RAAS and cardiac alterations have not been clearly elucidated. Therefore, the aims of this study were to investigate the alterations of cardiac autonomic activity, cardiac structure and RAAS in male rats after exposure to 4 weeks of CMS. We hypothesized that CMS could alter cardiac autonomic activity as reflected by HRV, in association with RAAS and cardiac hypertrophy.

## 2. Materials and Methods

### 2.1. Animals

Adult male Sprague Dawley rats (age, 7–8 weeks; weight, 240–250 g) were obtained from Nomura Siam International, BKK, Thailand. Upon arrival, all animals were housed in pairs in a standard individual ventilated cage with a cage divider and acclimatized for 7 days. The cage divider was used to limit direct physical contact between rats, but social stimulations from sound, sight, smell and minimal body contact were still possible. This divider also allowed data collection from individual rats [32]. All rats had free access to standard rat chow and water except in the case of some CMS procedures that required food or water deprivation. Rats were randomly divided into 2 groups: control group (*n* = 8) and CMS group (*n* = 8). To prevent the disturbance from CMS protocol to control animals, the CMS group was housed in a separate experimental room. The animal facility was maintained with a 12:12 h light/dark cycle (light on at 06:00 h) and under standard conditions of temperature (22 ± 1 °C) and humidity (50 ± 20%). All the experimental procedures in this study were carried out in accordance with the guidance for care and use of laboratory animals and approved by the Animal Care and Use Committee of Chulalongkorn University Laboratory Animal Center (protocol number 1773009).

### 2.2. Experimental Procedures

After acclimatization, all the rats were trained to wear a custom-made elastic cotton jacket (10 min/day, 7 days) for continuous electrocardiogram (ECG) recording. Thereafter, CMS rats were subjected to mild stressors for 4 weeks, while control rats were left undisturbed in their home cages. The ECG recording and sucrose intake were performed before starting CMS, and then every week to assess the cardiac autonomic nervous system activity and emotional reactivity, respectively. Body weight (BW) and food intake were recorded daily. At the end of the experiment, all rats were euthanized with an overdose of isoflurane (Terrell^TM^, Minrad, Bethlehem, PA, USA). Blood and tissues including the heart, lung, thymus gland and adrenal gland were collected, weighed and appropriately stored for subsequent analyses (Figure 1).

### 2.3. Chronic Mild Stress Procedure

The CMS procedure consisted of weekly exposure to food or water deprivation, group housing (4 rats/cage), a wetted cage, a 45°-tilted cage and continuous light (36 h), as previously described by Grønli et al. [33]. The CMS was performed for 4 weeks with fixed stressors every day (for details of daily stressors see Appendix A).

### 2.4. ECG Recording and HRV Measurement

The ECG was recorded using a custom-made elastic cotton jacket coupled with electrodes [34]. The jacket was customized so that the transthoracic positive and negative ECG could be recorded, and ground electrodes were positioned on the rat’s chest and back, respectively. The 10 min continuous ECG recording was performed between 09:00 and 11:00 h with a BIOPAC MP150 data acquisition system (Biopac Systems, Goleta, CA, USA). The recordings were manually edited and analyzed with LabChart 8 (ADInstruments, Castle Hill, NSW, Austrilia). The data from the best five consecutive min (i.e., clear tracing, no moving artifact) were used for analysis. The HRV analyses comprised both time and frequency domains. The time domain parameters were derived from the RR interval comprising the standard deviation of all the R–R intervals (SDNN), root mean square of successive differences between normal heartbeats (RMSSD) and percentage of subsequent R-R intervals with differences of duration longer than 10 ms (PNN10). The SDNN reflects the total variability. RMSSD and PNN10 are markers of parasympathetic activity [35]. For frequency domain parameters, the signals were filtered through a Hanning window and transformed into a spectrum by fast Fourier transformation. HRV parameters were analyzed from 512 samples of consecutive RR intervals [36]. In this study, the power spectrum consists of frequency bands of <0.2 Hz (very low frequency, VLF) [37], 0.20–0.78 Hz (low frequency, LF) and 0.78–2.5 Hz (high frequency, HF) [38,39]. The VLF, LF and HF represent the renin–angiotensin system, and sympathetic and parasympathetic activity, respectively. The ratio of LF to HF indicates the relationship between sympathetic and parasympathetic nerve activities (the sympatho-vagal balance). Upon data analysis, one rat from the control group and one from the CMS group were excluded from the experiment as an abnormality in ECG pattern, i.e., premature ventricular contraction (PVC), was evident.

### 2.5. Sucrose Intake Test

Sucrose intake was measured before the start of CMS as a baseline consumption (day −7 and day 0), and then once a week until the end of CMS [33]. All rats were restrictedfrom consuming food and water for 4 h (10:00–14:00 h), then the bottle containing 1% sucrose solution was provided for 1 h (14:00–15:00 h). Consumption was measured by comparing bottle weight before and after sucrose intake and normalized to BW. The higher sucrose intake indicated negative emotional reactivity-like stress.

### 2.6. Open-Field Test

The animals were tested in an open field one day after the end of CMS. The open-field test was conducted in accordance with McCarthy et al. (1995) [40]. The apparatus was a black wooden box (76 cm long × 57 cm wide × 50 cm high) with a 48-square grid floor (6 × 8 squares, 9.5 cm per side). The inner zone was defined as the innermost 8 squares, while the rest were defined as the outer zone. This test was conducted between 09:00 and 12:00 h in a dimly illuminated room with a light intensity of approximately 180–200 lx in the center of the apparatus. The experiments were recorded using a digital camcorder for subsequent analysis. Between each rat, the maze was carefully wiped with a wet towel and 70% ethanol and permitted to dry in between to remove olfactory cues [41]. The total number of lines crossed during 5 min in this task were counted and interpreted as the locomotor activity. The time spent in the outer or inner zone was counted. An increase in time spent in the outer zone or decrease in time spent in the inner zone indicated anxiety-like behavior.

### 2.7. Histomorphometric Analysis

Following organ weighing, each heart was first cut transversely at the level of the atrioventricular junction, and a second cut was made to obtain a heart ring of 2 mm thickness. The interventricular septum, and the right- and left-ventricular free wall were measured with an electronic caliper. Then, the heart sections were fixed with 10% neutral formalin. The formalin-fixed cardiac tissues were later embedded individually in paraffin. Transverse sections were cut at 4 µm thickness and stained with hematoxylin and eosin (H&E) according to standard protocol. The photomicrographs were taken under the light microscope at 40× magnification with a Canon E0S 550D camera, and the cardiomyocyte cross-sectional area was measured using a computer morphometry program (IMT i-Solution, Vancouver, BC, Canada). The cardiomyocyte was selected and measured only if the cell had a circular shape with a visible nucleus. For each rat, the cardiomyocyte cross-sectional area was measured and averaged from a total of 70 randomly selected cells.

### 2.8. Left-Ventricular ANP, BNP, AT1R, AT2R, ACE and ACE2 mRNA Expressions

On the day of euthanasia, the rest of the left-ventricular free wall was isolated, placed in RNAlater (QIAGEN, Hilden, NRW, DE) and stored at −80 °C. An RNA was isolated and transcribed into cDNA using the Aurum^TM^ Total RNA Mini Kit (BioRad, Hercules, CA, USA) and the iScript^TM^ Select cDNA Synthesis Kit (BioRad). Real-time PCR assays were performed in duplicate using the Real-Time-PCR Master Mix E4 kit (GeneON, Ludwigshafen, RP, Germany) with ABI PRISM^®^ 7700 instrument (Applied Biosystems, Foster City, CA, USA). The PCR reactions were performed under the following conditions: 95 °C (3 min) followed by 40 cycles of denaturation at 95 °C (30 s) with annealing and extension at 58 °C (1 min). To confirm product specificity, a dissociation step was performed at the end of PCR. Data were excluded if the dissociation curve had more than one peak. The relative amount of each target mRNA was calculated using the comparative threshold (CT) method [42]. Then, the relative mRNA expression was presented as 2^−∆CT^, where ∆CT was the difference between the CT of the target gene and of the internal control, the β actin. The nucleotide sequences of the primers used in this study are shown in Table 1.

### 2.9. Plasma B-Type Natriuretic Peptide (BNP) and Serum Ang II Concentration Measurements

Blood was collected by cardiac puncture during euthanasia, placed in microcentrifuge tubes with or without heparin, and centrifuged at 1000× *g* for 10 min at 4 °C. The plasma or serum was then separated and stored at −80 °C. The plasma BNP was measured using enzyme immunoassay test kit (RAB0386 Sigma-Aldrich, Saint Louis, MO, USA) with a lower/cv of 0.1 pg/mL/ < 10% and expressed as ng/mL. The serum Ang II was measured using an enzyme test kit (RAB0010, Sigma-Aldrich) with a lower/cv of 0.1 pg/mL/ < 10% and expressed as pg/mL.

### 2.10. Statistical Analysis

All data are presented as mean ± standard error of the mean (SEM). Food intake during the CMS period was recorded daily, normalized to BW, averaged and presented as average daily food intake (g/100 g BW). Time-related data were analyzed with two-way repeated measures Analysis of Variance followed by a Dunnett’s post hoc test; however, due to the differences at baseline, HRV parameters were first normalized to baseline to minimize individual differences; then, the effect of CMS was compared to the control using Student’s unpaired *t*-test at each time point. Parameters measured at the end of CMS, including relative organ weights (normalized to 100 g BW), open-field test, histomorphometry, relative mRNA expressions of interest genes and plasma measurements, were compared between groups using Student’s unpaired *t*-test. The correlations between parameters of interest were analyzed with the Pearson correlation. Differences were considered statistically significant at *p* < 0.05.

## 3. Results

### 3.1. Effects of CMS on Body Weight, Food Intake and Organ Weights

The body weight (BW) was affected by CMS (F_(1, 69)_ = 14.91, *p* = 0.002), days after CMS (F_(4, 69)_ = 445.78, *p* < 0.001) and interactions between factors (F_(4, 69)_ = 24.31, *p* < 0.001). The between groups comparison at each time point revealed that CMS had significantly lower BW than its control counterparts starting from the first week of CMS and remained significantly lower until the end of CMS. The within group comparison demonstrated that the BW were significantly increased from the first week and remained significantly higher until the end of CMS in both control and CMS groups (Figure 2A). For daily food intake, despite the food or water deprivation session of the CMS protocol, the average daily food intake was not different between groups (Figure 2B). The daily food intake during the 28 days of the CMS period is shown in Appendix A. For the relative organ weights (g/100 g BW) of control and CMS, the thymus gland was significantly decreased (0.15 ± 0.01 vs. 0.10 ± 0.02; *p* < 0.05), while the heart weight was significantly increased (0.27 ± 0.01 vs. 0.30 ± 0.01; *p* < 0.05). The lung and adrenal gland weights were not different (0.33 ± 0.01 vs. 0.33 ± 0.02 g/100 g BW and 12.00 ± 0.60 vs. 12.73 ± 0.56 mg/100 g BW, respectively; *p* > 0.05).

### 3.2. Effects of CMS on Emotional State

Sucrose intake was affected by CMS (F_(1, 82)_ = 9.87, *p* = 0.008), days after CMS (F_(5, 82)_ = 4.99, *p* < 0.001) and interactions between factors (F_(5, 82)_ = 3.62, *p* = 0.006). The between groups comparison at each time point revealed that sucrose intake of CMS was significantly higher than control at week 1, 2 and 4. The within group comparison demonstrated that in the control group, sucrose intake at week 2 (*p* < 0.01) and week 3 (*p* < 0.01) were significantly lower than baseline (week −1) as shown in Figure 3. For the CMS group, there were no significantly different from baseline. The non-normalized sucrose intake is shown in Appendix A. There were no behavioral differences in the open field between groups at the end of the CMS period (data not shown; *p* > 0.05).

### 3.3. Effects of CMS on Cardiac Autonomic Activity

Throughout the CMS period, the RR interval and HR were not significantly different between groups; however, it should be noted that the RR interval and HR of the CMS were likely to be lower and higher toward the end of CMS (Table 2). Due to individual differences, the CMS effect on HRV parameters was not significant as analyzed by Two-way repeated measure ANOVA; to minimize the differences, data were normalized to baseline (non-normalized data were shown in Appendix A).

For the time domain of HRV analysis including SDNN and RMSSD, the data were normalized to baseline (Table 2). The nSDNNs of CMS during the first 3 weeks were not different from the control (*p* > 0.05). However, at the end of CMS, the CMS rats had lower nSDNN than the control rats (*p* < 0.05). For nRMSSD, it was likely to be higher in CMS, but was not statistically different (*p* > 0.05).

For the frequency domain of HRV including VLF, LF and HF powers, and LF/HF ratio, the data were normalized to baseline (Figure 4). During the first 3 weeks, nVLF was not significantly different between groups; however, at the end of CMS, the CMS had significantly lower nVLF than that of the control (*p* < 0.01, Figure 4A). For nLF, the CMS had a higher value than the control, starting as early as the first week of CMS, and reached statistical significance at week 4 of CMS (*p* < 0.05, Figure 4B). For nHF, there was no difference between groups at the first 3 weeks of CMS, but the CMS had a significantly higher value than the control at week 4 of CMS (*p* < 0.05, Figure 4C). For the LF/HF ratio, there was no statistical difference between groups throughout the CMS period (Figure 4D).

### 3.4. Effects of CMS on Cardiac- and RAAS-Related Parameters

The relative heart weight was significantly higher in the CMS (0.27 ± 0.01 vs. 0.30 ± 0.01 g/100 g BW; *p* < 0.05), without significant changes in the right-, left- or inter-ventricular wall thicknesses (1.65 ± 0.05 vs. 1.68 ± 0.05, 3.76 ± 0.19 vs. 3.57 ± 0.18 and 2.70 ± 0.11 vs. 2.64 ± 0.15 mm, respectively; *p* > 0.05). Interestingly, the left-ventricular cardiomyocyte cross-sectional area of the CMS tended to be larger than that of the control (301.94 ± 23.73 vs. 346.75 ± 13.08 µm^2^; *p* = 0.062; Figure 5). For the relative mRNA expressions of ventricular ANP and BNP, there was no difference between groups (0.05 ± 0.03 vs. 0.04 ± 0.01 and 1.43 ± 0.28 vs. 1.19 ± 0.36, respectively; *p* > 0.05). Consequently, the plasma BNP was not significantly different between groups (0.75 ± 0.11 vs. 0.58 ± 0.14 ng/mL; *p* > 0.05).

For RAAS, the data revealed no significant difference between groups for the serum Ang II concentration (*p* > 0.05; Figure 6A) or the relative mRNA expressions of angiotensin-converting enzymes (ACE and ACE2) in the left ventricle (*p* > 0.05; Figure 6B). However, the relative mRNA expression of the angiotensin receptor AT1R was significantly increased in CMS (*p* < 0.05; Figure 6C), while the AT2R was not significantly different (*p* > 0.05; Figure 6C).

### 3.5. The Correlation between Stress-, HRV-, Cardiac- and RAAS-Related Parameters

Stress-related parameters including BW, relative adrenal gland weight and sucrose consumption were significantly correlated (*p* < 0.05, Figure 7). BW was negatively correlated with both adrenal gland weight and sucrose consumption, while adrenal gland weight was positively correlated with sucrose consumption. The thymus gland weight was excluded from further correlative analysis as it showed no correlation with other stress-related parameters (data not shown). Accordingly, the stress-related parameters were then correlated with other parameters. The correlations between stress- and HRV-related parameters were negative between BW vs. LF (*p* < 0.10) or HF (*p* < 0.05) and adrenal gland weight or sucrose consumption vs. LF/HF ratio (*p* < 0.05). Furthermore, sucrose consumption was positively correlated with HF (r +0.74, *p* < 0.05) but negatively correlated with SDNN (r −0.61, *p* < 0.05). In terms of cardiac structure, the heart weight was negatively correlated with BW (r −0.81, *p* < 0.05); however, it was positively correlated with sucrose consumption (*p* < 0.05) and tended to correlate with adrenal gland weight (*p* = 0.0527). For RAAS, the AT1R was negatively and positively correlated with BW and sucrose consumption, respectively (*p* < 0.05, Figure 7).

For HRV parameters, significant correlations were found within the frequency domain. Interestingly, LF was positively correlated with HF power (*p* < 0.05, Figure 7) and tended to positively correlate with LF/HF ratio (*p* = 0.0935, Figure 7), while HF was negatively correlated with LF/HF ratio (*p* < 0.05, Figure 7). Moreover, LF was positively correlated with cardiomyocyte cross-sectional area (*p* < 0.05). For LF/HF ratio, it was negatively correlated with serum angiotensin II or AT2R mRNA expression. For ANP and BNP mRNA expressions, both were negatively correlated with heart weight (*p* < 0.05).

As for RAAS, VLF power was positively correlated with BW but negatively correlated with sucrose consumption (*p* < 0.05). In relation to cardiac parameters, the data revealed negative correlations between VLF vs. heart weight (*p* < 0.10) and cardiomyocyte cross-sectional area (*p* < 0.05). Interestingly, the VLF was negatively correlated with all RAAS-related parameters, including serum angiotensin II, and mRNA expressions of AT1R, AT2R, ACE and ACE2; however, significant effects were found only for AT1R and ACE2 (*p* < 0.10). Similar to VLF, the serum angiotensin II and AT1R mRNA expression were positively correlated with heart weight (*p* < 0.05). On the other hand, AT2R was negatively correlated with cardiomyocyte cross-sectional area (*p* < 0.10).

## 4. Discussion

The present study investigated the impact of a 4-week CMS on cardiac autonomic activity, cardiac hypertrophy and RAAS-related parameters. Our data indicated that there was an alteration in both sympathetic and parasympathetic activities without affecting their balance, featuring mild cardiac hypertrophy with little or no effect on ANP/BNP or RAAS. At the end of CMS, the body weight of the CMS was 1.20-fold lower than the control. Despite the fact that the CMS protocol included food or water deprivation, the average daily food intake was not different between groups. This finding could be explained by the observation that the CMS increased food consumption following the deprivation sessions (Appendix A). It is thus likely that stress had a negative effect on body weight, and this is in agreement with previous studies [16,48,49,50,51,52]. As the weights of the adrenal gland and thymus gland could be affected by stress, they were used in this study as indirect stress indicators [53]. We found that CMS decreased thymus weight (1.42-fold), but the adrenal gland weight was not affected. The reduction in thymus gland weight was similar to previous studies [13,54,55]. With regard to the adrenal gland, changes in adrenal gland weight following CMS were shown to be increased [13,49,50,56] or unaffected [48,51,54]. The different outcomes could be explained by different stress protocols in terms of types and/or duration of stress or the time of measurement after stress cessation [48].

Sucrose intake was used in this study to assess animal reactivity to stress. We found that the CMS consumed a higher amount of sucrose. In stress animals, sucrose consumption usually increases [57,58], and it would decrease if the animal were depressed, known as the anhedonic effect. This finding indicated that the CMS group was stressed but had not yet developed into a depressive state [18,19]. Interestingly, the correlations between the stress-related parameters indicated that, the higher the stress level, as demonstrated by higher sucrose intake, the lower the body weight and the higher the adrenal gland weight.

Cardiac autonomic activity as shown by HRV analyses revealed that the SDNN was 1.5-fold lower in CMS. This was consistent with previous studies [14,15,16,17] and indicated that CMS reduced overall HRV. We also found that the SDNN was negatively correlated with sucrose intake, which suggested that the higher the stress response, the lower the HRV. In terms of the frequency domain of HRV, LF and HF, the indicators of sympathetic and parasympathetic activities had both increased (1.9- and 2.0-fold, respectively) at the end of CMS. This suggested that cardiac autonomic activity increased. It is worth mentioning that LF increased along with the stress duration, reaching significance at the end of CMS. The increment of LF was positively correlated with that of HF, and these increments resulted in an unaltered LF/HF ratio. The increased LF power following stress was reported by many previous studies utilizing various stress models including psychological stress [59], social defeat stress [13,52] and the CMS model [16,17]. On the contrary, most studies showed that the HF power was either unchanged [60] or decreased [16,17,52]. Therefore, the increase in HF power at 4 weeks in this study was unexpected. In this study, LF/HF ratio was unaffected, and it was correlated with both LF and HF, which suggested that the increased parasympathetic activity served to counterbalance the increased sympathetic activity. HRV analysis indicated that the 4-week CMS increased cardiac autonomic activity without affecting autonomic balance.

Cardiac hypertrophy is an adaptive response of the heart to physiological and pathological stimuli, and commonly manifests as cardiomyocyte enlargement, increased wall thickness and/or increased cardiac mass [20]. In the current study, we found a mild increase in heart weight (10.74 ± 2.84%) with the possibility of a larger cardiomyocyte cross-sectional area (1.15-fold, *p* = 0.062). Despite these cardiac alterations, the ventricular mRNA expressions of ANP and BNP, and plasma BNP were not affected. However, the negative correlations between heart weight vs. ANP and BNP mRNA expressions were evident. The NPs are clinically and experimentally used as biomarkers for hypertrophy and heart failure; their levels appeared to increase during cardiac changes [28]. Thus, one would expect to see negative correlations between NPs and hypertrophic parameters; however, NPs mRNA level only increased during fibrosis development of late-stage hypertrophy [28]. Accordingly, it suggests that CMS-induced cardiac hypertrophy in this study is mild and in an adaptive, but not yet pathological, stage.

In terms of RAAS-related activity, the lower VLF and the increased ventricular mRNA expression of AT1R were demonstrated in the CMS. The lower VLF of the HRV frequency domain can be explained by the higher activity of RAAS, as it was shown experimentally that the higher the RAAS activation, the lower the VLF power [26]. This relationship was strengthened by correlative analyses, from which we found negative correlations between VLF and all RAAS-related parameters, despite the significance. Overall, it was implied that the RAAS was activated in stressful conditions. Unexpectedly, the serum Ang II was not increased in CMS rats, which is probably due to the very short half-life of Ang II (14.8 s) [61].

In order to establish that CMS induced cardiac hypertrophy through sympathetic and/or RAAS activation, correlations between stress-, HRV-, cardiac- and RAAS-related parameters were performed. We found that the stress-related parameter was correlated with SDNN, LF, heart weight and AT1R mRNA expression, indicating that stress could lead to sympathetic and RAAS activations, and cardiac hypertrophy. Correlations were also established between the LF or VLF and cardiomyocyte cross-sectional area, which suggested that sympathetic and RAAS activations were responsible for cardiac hypertrophy. Moreover, a correlation between LF and VLF as an indicator of sympathetic and RAAS activity was established. Therefore, it was likely that these two major regulatory systems interacted with each other and contributed to the genesis of cardiac hypertrophy [62]. The mechanism of sympathetic-induced cardiac hypertrophy could be explained by in vivo and in vitro studies, in that NE could induce cardiomyocyte hypertrophy and apoptosis, and cardiac fibroblasts proliferation and collagen expression [8,22]. RAAS-induced cardiac hypertrophy could occur through the classic RAAS pathway (ACE/AngII/AT1R) while being preventing by the new RAAS pathway (ACE2/Ang1–7/Mas1R and AT2R) [23,24,25]. This is interesting because, in this study, the AT1R mRNA was upregulated and was positively correlated with heart weight, suggesting its hypertrophic role; on the other hand, a negative correlation was found between AT2R and cardiomyocyte cross-sectional area, suggesting its protective effect. In summary, 4-week CMS in male rats induced negative emotion as shown by stress-related parameters, and increased cardiac autonomic and RAAS activities which may be responsible for mild cardiac hypertrophy. The cardiac hypertrophy herein was postulated to be in an adaptive, not pathological, stage and the cardiac autonomic function was preserved as the sympathetic and parasympathetic activities were in balance.

One limitation of this study was the use of mRNA rather than protein expressions as indicators; in some cases, the increased mRNA may not be accompanied by increased protein expressions. However, a good agreement is generally established between the mRNA and protein expressions. Another limitation was the fact that changes in cardiac, autonomic and RAAS activities following CMS were based on correlations and could not exclusively be proven as all the parameters were indeed collected at the end of the CMS period, and no sequential analysis was performed. Additionally, it should be noted that, the longer the stress, the higher the sympathetic function; therefore, changes in cardiac autonomic control and cardiac structure might have been more significant if the CMS had been prolonged.

## 5. Conclusions

At the end of 4 weeks of CMS, HRV changed, manifesting as a decreased SDNN value, increased LF power and, surprisingly, increased HF power. As a result, the LF/HF was unchanged, indicating that CMS for 4 weeks does not affect cardiac autonomic balance. At the same time, CMS induced cardiac hypertrophy via sympathetic and RAAS activation. These cardiac alterations were likely not in a pathological stage as it was not accompanied by any changes in ANP and BNP mRNA expressions, and the cardiac autonomic activity was still in balance. This suggests that cardiac alterations during CMS are a primary adaptive mechanism against cardiac autonomic changes.

## Figures and Tables

**Figure 1 vetsci-09-00539-f001:**
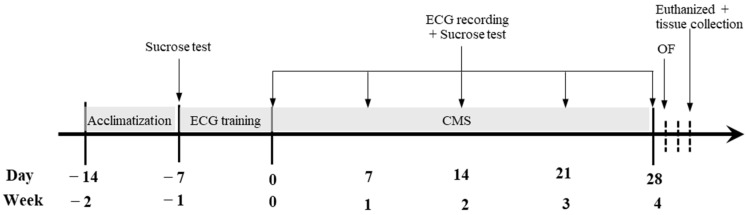
Overview of the experiment time course for CMS and ECG recording.

**Figure 2 vetsci-09-00539-f002:**
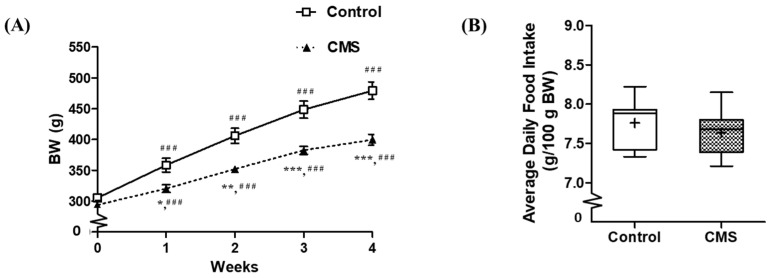
Body weight (**A**) and average daily food intake (**B**) of control and CMS. * *p* < 0.05; ** *p* < 0.01; *** *p* < 0.001 vs. control at the same time point, ^###^
*p* < 0.001 vs. baseline (week 0) within group, two-way repeated measures ANOVA followed by Dunnett’s; *n* = 7 for each group.

**Figure 3 vetsci-09-00539-f003:**
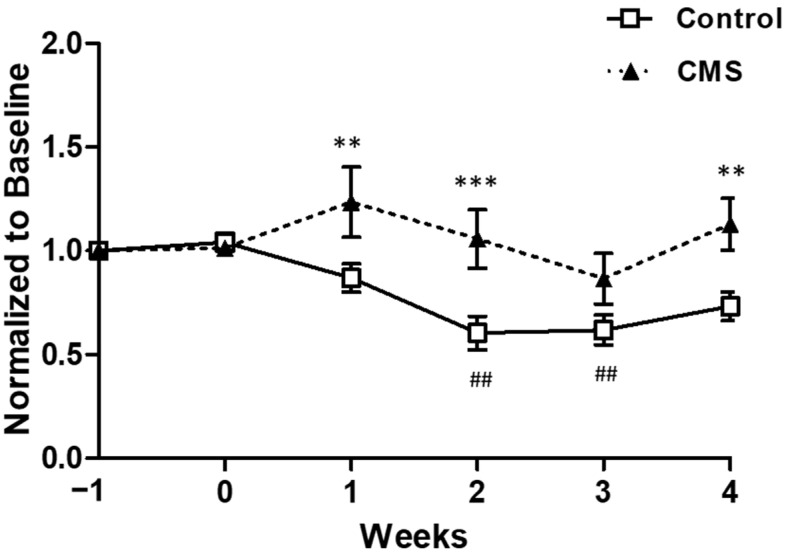
Sucrose intake of control and CMS. Data are presented as mean ± S.E.M. ** *p* < 0.01, *** *p* < 0.001 vs. control at the same time point, ^##^
*p* < 0.01 vs. baseline (week −1) within group, two-way repeated measures ANOVA followed by Dunnett’s, *n* = 7 for each group.

**Figure 4 vetsci-09-00539-f004:**
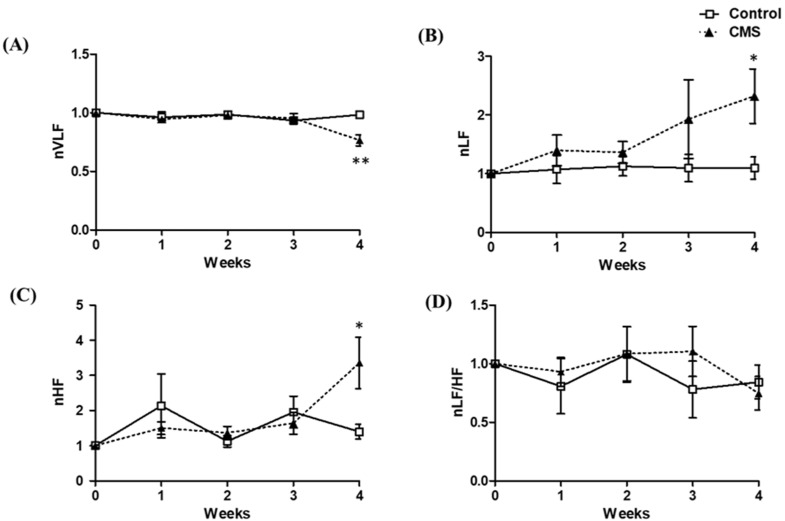
Normalized data of VLF (**A**), LF (**B**), HF (**C**) and LF/HF ratio (**D**) of control and CMS. Data are presented as mean + S.E.M., * *p* < 0.05 vs. control, ** *p* < 0.01 vs. control at the same time point, Student’s unpaired *t*-test; *n* = 7 for each group.

**Figure 5 vetsci-09-00539-f005:**
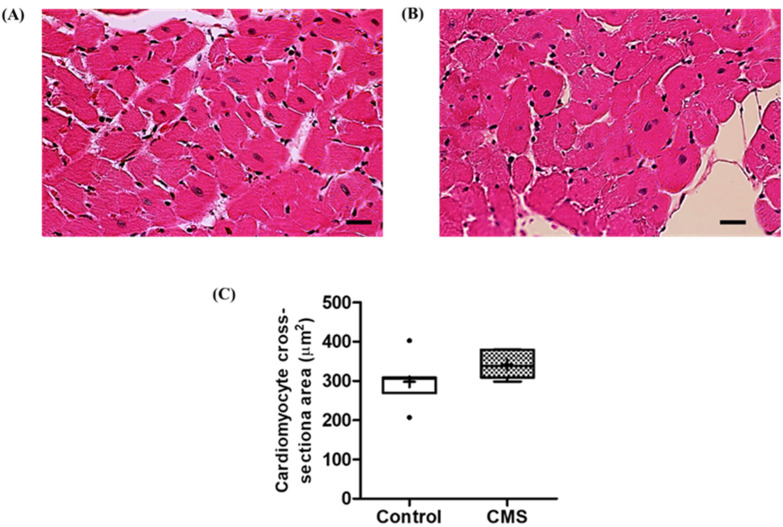
Histological sections of left ventricle in control (**A**) and CMS (**B**), and box plot of quantitative analysis of cardiomyocyte cross-sectional area (**C**). H&E, 400×; scale bar = 20 µm; *n* = 7 for each group.

**Figure 6 vetsci-09-00539-f006:**
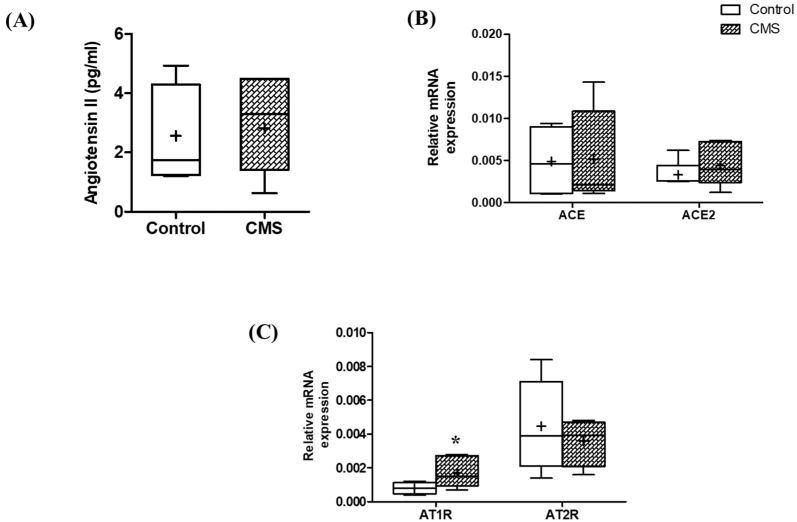
Box plots of RAAS including Ang II concentration (**A**), angiotensin-converting enzymes (ACE and ACE2) (**B**) and angiotensin receptors (AT1R and AT2R) (**C**) in control and CMS groups. * *p* < 0.05 vs. control using Student’s unpaired *t*-test; *n* = 4–7 for each group.

**Figure 7 vetsci-09-00539-f007:**
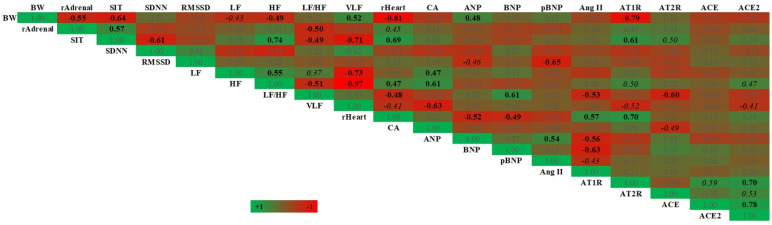
Correlative analysis between stress-, HRV-, cardiac- and RAAS-related indices. The correlative values are shown with the gradient color of green to red ranging from +1 to −1; numbers in bold and italic represent *p* < 0.05 and 0.05 < *p* < 0.10, as determined by Pearson correlation; *n* = 13–14. Abbreviations: BW, body weight; CA, cardiomyocyte area; pBNP, plasma BNP; rAdrenal, relative adrenal gland weight; rHeart, relative heart weight; SIT, normalized sucrose intake test.

**Table 1 vetsci-09-00539-t001:** Nucleotide sequences of primers used in real-time PCR.

Gene	Forward Primer	Reverse Primer	References
ANP	GAGGAGAAGATGCCGGTAG	CTAGAGAGGGAGCTAAGTG	[43]
BNP	TGATTCTGCTCCTGCTTTTC	GTGGATTGTTCTGGAGACTG	[43]
AT1R	TCTGGATAAATCACACAACCCTC	GAGTTGGTCTCAGACACTATTCG	[44]
AT2R	CTGGCAAGCATCTTATGTAGTTC	ACAAGCATTCACACCTAAGTATTC	[44]
ACE	TCCTATTCCCGCTCATCT	CCAGCCCTTCTGTACCATT	[45]
ACE2	GAGGAGAATGCCCAAAAGATGA	GAAATTTTGGGCGATCTTGGA	[46]
β-actin	AGGGAAATCGTGCGTGAC	CGCTCATTGCCGATAGTG	[47]

**Table 2 vetsci-09-00539-t002:** RR interval, HR, normalized SDNN and RMSSD of control and CMS.

Parameters	Week 0	Week 1	Week 2	Week 3	Week 4
RR interval (ms)					
Control	124.39 ± 2.62	122.69 ± 2.41	129.47 ± 2.92	132.24 ± 4.07	132.14 ± 5.88
CMS	118.86 ± 3.23	124.49 ± 4.26	127.64 ± 4.15	131.43 ± 3.27	123.99 ± 2.00
HR (bpm)					
Control	484.80 ± 9.71	491.06 ± 9.88	467.00 ± 9.99	456.97 ± 13.04	460.47 ± 19.20
CMS	508.11 ± 13.07	473.86 ± 14.96	473.86 ± 14.96	459.64 ± 11.50	485.29 ± 7.91
nSDNN					
Control	1	0.85 ± 0.05	1.19 ± 0.14	0.98 ± 0.15	1.14 ± 0.12
CMS	1	1.04 ± 0.17	1.01 ± 0.14	1.26 ± 0.17	0.83 ± 0.10 *
nRMSSD					
Control	1	1.08 ± 0.17	1.01 ± 0.16	1.27 ± 0.23	1.08 ± 0.15
CMS	1	1.09 ± 0.08	1.10 ± 0.08	1.38 ± 0.09	1.39 ± 0.25

Data are presented as mean ± S.E.M., * *p* < 0.05 vs. control at the same time point, Student’s unpaired *t*-test; *n* = 7 for each group.

## Data Availability

The data supporting the findings of this study are available within the article.

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
