# Peer review of "Effects of Chronic Mild Stress on Cardiac Autonomic Activity, Cardiac Structure and Renin–Angiotensin–Aldosterone System in Male Rats"

_vetsci, 2022, doi:10.3390/vetsci9100539_

Round 1
Reviewer 1 Report
The autonomic imbalance was previously determined to be associated with stress and further cardiovascular diseases. In the provided study, the authors were investigated effects of chronic mild stress (CMS) on cardiac autonomic control, cardiac structure, and RAAS activity in adult male Sprague-Dawley rats. It was suggested that that 4-week CMS in male rats induced negative emotion as shown by increased sucrose intake and increased cardiac autonomic and RAAS activities, which may be responsible for mild cardiac hypertrophy. Moreover, the cardiac autonomic function was preserved as the autonomic activities were balanced.
The authors provided well written paper where the scientific problem has been adequately postulated, with methodological approach, statistical analysis and required ethical procedure properly executed. The discussion section covered all relevant knowledge important for this paper with adequate analysis considering the results of other scientists. The clinical significance of this study has been clearly underlined, as well as several limitations. All together, the obtained results will certainly help further expanding and clarifying the scientific problem of chronic mild stress.
Author Response
We would like to express our thanks for your valued comments.
Reviewer 2 Report
The chronic mild stress (CMS) protocol is not clearly outlined. When each week was each deprivation and when was sucrose available (were these separate vs concurrent)? For example, the authors mention that sucrose intake measured after after 4 hrs of food AND water deprivation (line 166). That statement implies that the restriction was concurrent and that 1% sucrose was available during deprivation. Intake in sucrose may be an indication of nutrient and water intake differences not CMS.
The authors present their findings as outcomes associated with Stress. However, the experimental mechanism by which stress is achieved (e.g. dehydration, increased sucrose intake) alter blood pressure parameters them selves (like changes in osmotic pressure and blood viscosity). The authors did not provide direct evidence that these animals were “stressed”. Nor can the authors differentiate the outcomes as independent of the hemodynamic effects of their CMS protocol.
The results of this study are tenuous at best but do not support the study aim.
It is also particularly disappointing when so many parameters are being compared and analyzed but so little information is provided in the statistical analysis section. This needs to be expanded.
The authors should include original data if they plan to present normalized data. I would suggest in table form with statistical results from that raw data. This can be supplemented with “normalized data” in figures that do not include trend lines because the rate of change cannot be assumed between time points (figures 3,4).
Bar graphs should be replaced with box plots to better demonstrate the distribution of data (figures 2, 5, 6.
Any experiment with repeated measures (like weight and sucrose intake) should not be analyzed with unpaired statistics. These should be completed using a repeated measures ANOVA with a post-hoc comparison test. Lack of significance in these tests should be reported.
The authors did not make a compelling link between sucrose intake and “emotional state”
Authors make several claims throughout the manuscript that are not supported by the data. Once the stats are redone these should be revised for accuracy.
Isolated results at 4 weeks (e.g. nLF and nSDNN) are not supported with trends. It is therefore possible that these are anomalous responses based on the low sample size and high variability. These parameters should be repeated and/or assessed at 5 weeks to confirm response. If that is not the case then a statement on the limitations of the sample size And study duration should be included in the discussion.
Figure 6 indicates a sample size of 4-7. The variation in sample size must be clearly explained and number of animals per test described.
Statements are made throughout that require literature support including line 34, line 58, line 169, and line 183. Without literature support these should be omitted or revised.
Animal ages should be included.
Considerable English editing is required.
Figure 7 is illegible and must be revised.
References are not formatted consistently
Author Response
We would like to express our thanks for your valued comments and below are our responses.
The chronic mild stress (CMS) protocol is not clearly outlined. When each week was each deprivation and when was sucrose available (were these separate vs concurrent)? For example, the authors mention that sucrose intake measured after after 4 hrs of food AND water deprivation (line 166). That statement implies that the restriction was concurrent and that 1% sucrose was available during deprivation. Intake in sucrose may be an indication of nutrient and water intake differences not CMS.
The CMS protocol in detail was added as supplement material (Table S1). For sucrose intake test, the food and water deprivation before the test was not part of CMS protocol, the test was performed at the same time for both control and CMS groups when there was no stress given to CMS group. We have clarified this point in the section 2.5. Sucrose intake test.
The authors present their findings as outcomes associated with Stress. However, the experimental mechanism by which stress is achieved (e.g. dehydration, increased sucrose intake) alter blood pressure parameters themselves (like changes in osmotic pressure and blood viscosity). The authors did not provide direct evidence that these animals were “stressed”. Nor can the authors differentiate the outcomes as independent of the hemodynamic effects of their CMS protocol.
In term of stress, we stated that exposed to 4-week CMS reduced body weight, increased sucrose consumption, and reduced thymus gland weight, these parameters were also used as stress indicators in literatures. Accordingly, we stated that the animals were stress, and these assumptions were in the discussion section. For the comment above regarding the hemodynamic changes following CMS protocol such as water deprivation, we agreed with the reviewer that it was possible. Unfortunately, we did not have any parameters to clarify this point. However, according to experimental timeline, the weekly ECG recording was done before sucrose test and before the water deprivation of that week, therefore, these protocols should have less effect on HRV data if any. The more detail of CMS and sucrose intake was added in revised manuscript, this point should be then clarified.
The results of this study are tenuous at best but do not support the study aim.
The aims of the study were to determine the effect of CMS on cardiac autonomic control, structure and RAAS, the results shown that both sympathetic and parasympathetic activities were increased, the RAAS was also increased while the alteration of cardiac structure was not as signified. This issue was already addressed in the discussion.
It is also particularly disappointing when so many parameters are being compared and analyzed but so little information is provided in the statistical analysis section. This needs to be expanded.
This has been added to section 2.10. Statistical analysis as suggested.
The authors should include original data if they plan to present normalized data. I would suggest in table form with statistical results from that raw data. This can be supplemented with “normalized data” in figures that do not include trend lines because the rate of change cannot be assumed between time points (figures 3,4).
Any experiment with repeated measures (like weight and sucrose intake) should not be analyzed with unpaired statistics. These should be completed using a repeated measures ANOVA with a post-hoc comparison test. Lack of significance in these tests should be reported.
The original data were included as supplement materials, only normalized data were shown in the manuscript as stated in the statistical analysis section. Unexpectedly, we noticed that there were differences in baseline value despite the randomization of the rats to study groups, we therefore normalized data to baseline. This would eliminate the individual differences and thus better reflected CMS effect.
Bar graphs should be replaced with box plots to better demonstrate the distribution of data (figures 2, 5, 6.)
The bar graphs were already replaced by box plots as suggested.
The authors did not make a compelling link between sucrose intake and “emotional state”
In discussion we stated that “In stress animals, the sweet consumption was usually increased [55, 56], and it would be decreased if the animal was depressed, known as anhedonic effect.”, and therefore, the increased sucrose intake in CMS indicated that animals were stressed. This assumption was included in the discussion.
Authors make several claims throughout the manuscript that are not supported by the data. Once the stats are redone these should be revised for accuracy.
As answer in previous comment about statistical analysis, we have performed statistic as suggested; however, from our point of views, we would like to use the current analysis as it better represented CMS effects.
Isolated results at 4 weeks (e.g. nLF and nSDNN) are not supported with trends. It is therefore possible that these are anomalous responses based on the low sample size and high variability. These parameters should be repeated and/or assessed at 5 weeks to confirm response. If that is not the case then a statement on the limitations of the sample size And study duration should be included in the discussion.
The limitation on study duration was added in revised manuscript as suggested.
Figure 6 indicates a sample size of 4-7. The variation in sample size must be clearly explained and number of animals per test described.
As previously indicated in section 2.8. Left ventricular ANP, BNP, AT1R, AT2R, ACE and ACE2 mRNA expressions, that if the dissociation curve had more than 1 peaks, the specific data point would be excluded from analysis.
Statements are made throughout that require literature support including line 34, line 58, line 169, and line 183. Without literature support these should be omitted or revised.
We already added references to support these statements in revised manuscript.
Animal ages should be included.
The animal ages had already added to section 2.1 as suggested.
Considerable English editing is required.
The revised manuscript had been sent for English edited through MDPI as suggested.
Figure 7 is illegible and must be revised.
The figure was revised as suggested. Only the r2 reaching statistical significance or trended were shown and the font color was changed to make it easier to read. The figure also submitted as separated file for better view.
References are not formatted consistently
We have rechecked and made correction as in revised manuscript.
Round 2
Reviewer 2 Report
I would like to thank the authors for addressing many of the reviewer comments and suggestions to improve the quality of this manuscript.
The others include modified/normalized data in the manuscript. I asked the the authors include a table with raw data and results from analysis with that raw data. These have not been included (see previous comment below).
The authors should have performed RM-ANOVAS for data measured repeatedly rather than unpaired t-tests. The previous comment in also included below. Once this data and analysis is included and discussed in the manuscript the paper should be acceptable for publication.
"The authors should include original data if they plan to present normalized data. I would suggest in table form with statistical results from that raw data. This can be supplemented with “normalized data” in figures that do not include trend lines because the rate of change cannot be assumed between time points (figures 3,4).
Any experiment with repeated measures (like weight and sucrose intake) should not be analyzed with unpaired statistics. These should be completed using a repeated measures ANOVA with a post-hoc comparison test. Lack of significance in these tests should be reported."
Author Response
Dear Reviewer,
We would like to express our thanks for your comments and below are our responses.
According to your comments, we have added the statistical analysis in method, results, and supplementary materials as shown in revised manuscript.
Best regards,
SKT
